# Synergistic effects of physical-mental mixed fatigue on badminton forehand smash performance

Bing Wang[1☉], Lan Ding[2,3☉], Meng Liu[1]*, Yan Huang[2,4]*, Jun Ai[1]

1 School of Physical Education, University of Jinan, Jinan, China, 2 Sports Coaching College, Beijing Sport University, Beijing, China, 3 Beijing University of Posts and Telecommunications, Beijing, China, 4 Beijing Shichahai Sports School, Beijing, China

☉ These authors contributed equally to this work.
* spe_lium@ujn.edu.cn (ML); 13811386756@163.com (YH)

## Abstract

### Purpose

Badminton's forehand jump smash is critical for offensive success, yet its performance often declines under fatigue. Previous studies have separately examined physical or mental fatigue, neglecting their combined impact, common in competitive scenarios. This study investigates the synergistic effects of physical-mental mixed fatigue on smash performance, focusing on speed and accuracy in badminton athletes using a dual-channel neuromuscular-mental fatigue model.

### Method

Twenty-four national-level 1 badminton athletes participated. A fatigue protocol combining vertical jumps, sprints, and randomized target-hitting tasks induced simultaneous physical and mental fatigue. Athletes executed 10 smashes at baseline and after mild, moderate, and severe fatigue. Smash speed was measured by radar, and accuracy was scored via a camera-based system. Fatigue levels were assessed with RPE and VAFS scales, analyzed through repeated-measures ANOVA and linear mixed-effects models.

### Results

Increased fatigue significantly reduced smash speed by 10.6% and accuracy by 46.1% from baseline to severe fatigue conditions (both $p < 0.001$). Under moderate-to-severe fatigue, a significant negative correlation between smash speed and accuracy emerged (moderate fatigue $r = -0.58$, severe fatigue $r = -0.53$, both $p < 0.01$), indicating a notable speed-accuracy trade-off.

**Data availability statement:** All relevant data are within the paper and its Supporting information files.

**Funding:** This work was supported by the General Administration of Sport of China National Team Youth Science and Technology Support Project (No. 25QN011).

**Competing interests:** The authors have declared that no competing interests exist.

## Discussion

Physical-mental mixed fatigue significantly impairs badminton smash performance, particularly accuracy. The observed speed-accuracy trade-off highlights challenges in maintaining technical quality under fatigue. Future research should explore broader athlete populations and integrate neurophysiological measures, informing strategies for cognitive-physical training and fatigue management in elite badminton athletes. These findings suggest that coaches should integrate fatigue simulation and mitigation strategies into technical training to sustain performance under competitive conditions.

## Introduction

Badminton represents one of the world's most popular racquet sports, characterized by high-speed exchanges, rapid directional changes, and complex decision-making under time constraints [1]. At elite competitive levels, success depends not only on technical proficiency but also on maintaining performance quality under progressive fatigue. Among badminton's offensive techniques, the forehand smash stands as the most explosive and decisive, generating shuttlecock velocities exceeding 400 km/h and constituting approximately 20% of offensive actions in competitive matches [2,3]. Fatigue in sports manifests through distinct but interconnected pathways. Physical fatigue primarily results from metabolic perturbations and neuromuscular impairments, including energy substrate depletion, metabolite accumulation, and compromised excitation-contraction coupling [4]. In intermittent high-intensity activities like badminton, physical fatigue typically reduces power output, movement velocity, and biomechanical efficiency [5]. Concurrently, cognitive fatigue—induced by sustained attentional demands and complex decision-making—impairs perceptual-cognitive function through attentional narrowing, delayed information processing, and compromised executive function [6]. This cognitive deterioration manifests as decreased technical accuracy, delayed reaction time, and sub-optimal tactical decisions [7].

Recent research has documented fatigue-induced performance decrements in badminton-specific contexts. Le Mansec et al. (2020) demonstrated that muscular fatigue significantly reduces smash velocity and stroke height through impaired lower-limb power production. Similarly, Kosack et al. (2020) observed that mental fatigue compromises decision-making speed and attentional allocation during badminton tasks, leading to increased technical errors [8]. Other studies have identified fatigue-related alterations in movement patterns, including compromised lunge stability [9], diminished net-play precision [10], and deteriorated visual search strategies during anticipatory tasks.

Fatigue not only impairs technical performance but also alters neuromuscular control, increasing injury risk during explosive movements such as the forehand jump smash. Fatigue-induced changes in landing biomechanics can elevate anterior cruciate ligament (ACL) loading, and quadriceps fatigue compromises knee joint stability

during badminton-specific actions like scissor jumps [11,12]. These findings underscore the importance of understanding fatigue from both a performance and injury prevention perspective.

While we acknowledge the value of investigating physical and mental fatigue independently, this research addresses the more ecologically valid scenario in which both fatigue modalities co-occur. Therefore, we introduce a novel neuromuscular-cognitive dual-channel fatigue model, designed to simulate the combined physical and mental demands encountered in competitive badminton. This dual-channel model conceptually aligns with established fatigue theories such as the Central Governor Model and psychobiological models of fatigue, which emphasize the integrative control of motor output under combined physiological and cognitive stress [13,14]. Using a graded fatigue protocol and multidimensional assessment—including jump height, smash speed, and accuracy—we aim to quantify how increasing levels of mixed fatigue impact forehand jump smash performance. We hypothesize that the interaction of physical and cognitive fatigue will produce greater performance deterioration than either fatigue modality alone, potentially revealing critical thresholds and speed-accuracy trade-offs [15]. This hypothesis is supported by prior findings from Afzal et al. (2020), who demonstrated a significant speed-accuracy trade-off in elite badminton players during forehand smashes [16]. Their work highlights that higher shuttlecock speeds are often achieved at the expense of spatial precision, providing a conceptual and empirical foundation for our expectation of SAT effects under fatigue.

In actual competitive scenarios, athletes are rarely subjected to purely physical or mental fatigue in isolation. Instead, performance deterioration often arises from the interaction of both fatigue modalities. Therefore, this study focuses specifically on the combined effects of physical and mental fatigue, aiming to simulate ecologically valid fatigue conditions reflective of real-world gameplay.

## Methods

### Participants

Sample size determination was performed using G*Power 3.1.9.7 with parameters: Cohen's f effect size = 0.25, power = 0.8, significance level $\alpha$ = 0.05, and critical F-value = 2.74, yielding a minimum required sample size of 24 participants. Therefore, 24 badminton athletes (20 males, 4 females) meeting the following inclusion criteria were recruited: Certified as National Level 1 Athletes or above per China's Athlete Technical Grading Standards; no history of chronic fatigue syndrome or musculoskeletal injuries; aged 18–28 years (Table 1). For female participants, menstrual cycle information was collected via self-report. All tests were scheduled during the follicular phase (days 6–12) to minimize the potential influence of hormonal fluctuations on fatigue and performance variables. All participants provided written informed consent. The study protocol received approval from the Ethics Committee of Beijing Sport University (approval number 2023073H), in accordance with the Declaration of Helsinki. Participants were recruited from March 1, 2024, to April 1, 2024. The experimental procedures were conducted from April 10, 2024, to June 25, 2024.

### Study design

At the beginning of the meeting, all the participants were familiar with all the procedures. Before and after the completion of the fatigue protocol (see below for details), participants take a specific badminton test (see specific test) to measure the quality of forehand smash (accuracy and badminton speed).

**Table 1. Basic Characteristics of Participants.**

| N | Gender (male/female) | Age (years) | Height (cm) | Body mass (kg) | Training experience (years) |
|---|---|---|---|---|---|
| 24 | 20:4 | 21.76 ± 1.93 | 177.15 ± 7.05 | 71.71 ± 8.70 | 10.91 ± 2.38 |

## Specific test

The test protocol required participants to perform 10 full forehand spikes, aiming at two geometric zones (212×40 cm zone control area) on the sideline of singles. To standardize biomechanical comparability, male participants executed the forehand jump smash, while female participants performed a restricted-step stationary stroke. Due to well-documented differences in average jump height and power output between elite male and female athletes in our recruitment pool, the female participants performed the stationary stroke to ensure technical safety and movement standardization. While this limits direct biomechanical comparison, it allows for the assessment of within-subject fatigue effects on each athlete's maximal performance capability. The ball machine (EDIBO 3.0, launch frequency 3 Hz±0.5°, shuttlecock speed 70±2 m/s, coordinate positioning error < 1 cm) was positioned 75–80 cm behind the doubles service line, projecting shuttles toward the court center with ascending trajectories at 3-second intervals using new shuttlecocks for each trial (Fig 1). The performance assessment was explained to the participants before starting the test. During the test, participants were strongly encouraged and informed of their outcomes in order to maintain vigilance and concentration throughout the procedure.

## Physical and mental mixed fatigue

To induce physical and mental mixed fatigue, refer to the badminton body fatigue induction scheme of Yann Le Mansec [8] and improve it (Fig 2): The subjects complete 10 vertical jumps with their hands akimbo in 5 seconds, and then perform 15m sprint and 10 random shots in the whole court. The completion of these three activities is one group, 10 groups are one Block, and two blocks are one unit, each unit time is about 20 min. The criteria for judging fatigue degree are: completing Block1 and Block2 with mild fatigue, completing Block3 and Block4 with moderate fatigue, and t and testing Rating of Perceived Exertion (RPE) and Visual Analog Fatigue Scale (VAFS). Block5 and Block6 are

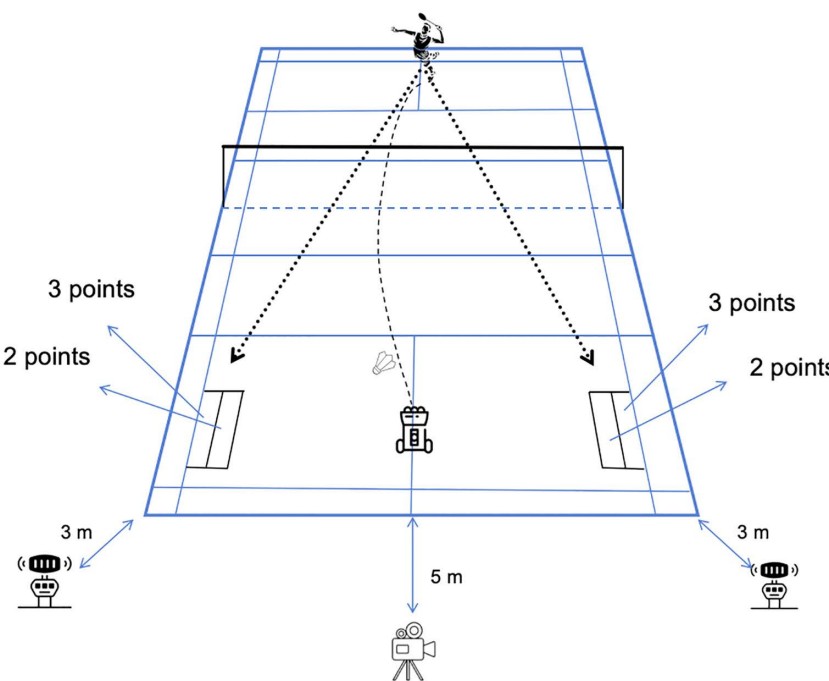

**Fig 1. Schematic diagram of the forehand smash test.**

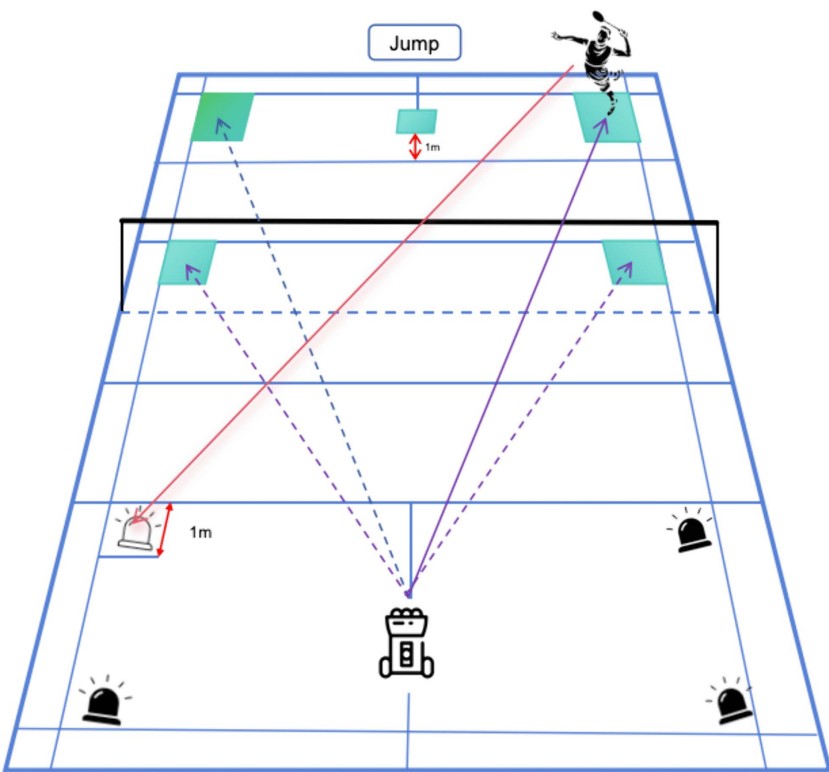

**Fig 2. Schematic Diagram of Randomized Target Hitting.**

completed due to severe fatigue. Immediately after each unit is completed, the forehand smash speed and accuracy are tested, RPE and VAFS tests are carried out. Mental fatigue is induced by randomly hitting the ball according to the signal light, so that the subjects can keep continuous attention during exercise and hit the ball according to the signal prompt. In this way, both physical fatigue and mental fatigue are induced. Instead of inducing mental fatigue through long-term mental activities such as cognitive tests. Random hitting is one of the four gray areas on the right randomly issued by the server. At the same time of serving, a signal light in the four gray areas on the left lights up randomly. Participants can use any badminton technique to hit the ball to the lighted area, and they need to complete 10 successful shots. After hitting the ball, the subjects return to the baseline position (vertical jump) and start a new cycle group. Test directly after the last group. During the test, the tester constantly encourages the athletes to make the subjects do their best to test and induce fatigue. During the fatigue induction protocol, the visual signal changed every 2–4 seconds in a randomized sequence among four predefined target zones, each with an equal probability of selection. This randomization was implemented using a computer-controlled system to prevent anticipatory responses and to simulate the unpredictable nature of in-game rallies. A successful shot was defined as one in which the shuttlecock landed within ±5 cm of the designated target area and was executed within 2 seconds following the presentation of the visual signal. The combined fatigue protocol was specifically developed to emulate the simultaneous physical and cognitive demands encountered in competitive badminton, incorporating elements such as high-intensity footwork, rapid and adaptive decision-making, and exposure to variable and unpredictable shuttle trajectories. These features were intended to closely approximate the dynamic and multifactorial nature of real match play, thereby strengthening the ecological validity of the experimental design.

## Data processing

**Shuttlecock speed.** Every forehand smash, the speed at which the players hit the badminton is measured by radar (Stalker ATS, USA). The radar is located 3 meters behind the players with a height of 2.50 meters. In order to ensure that the speed data is recorded, the experimenter manually points the radar at the area aimed at by the player. All the data are recorded on a personal laptop. The shuttlecock speed was measured for each of the 20 smashes under every fatigue condition (Baseline, Mild, Moderate, and Severe). For each condition, the average stroke speed was calculated by averaging 10 consecutive valid speed values, resulting in four mean speed values per participant (see Fig 1 for radar positioning).

**Precision.** The accuracy was evaluated by using a camera (AHD-H12 VAZ2S, Aiptek, Rowland, Street, Ca, USA) for post-processing analysis. The camera is located 5 meters behind the baseline opposite the player. A three-tiered scoring system was implemented with the following criteria: 3 points awarded for shots landing in the $\pm 5$ cm core zone adjacent to the sideline, 2 points for the $\pm 10$ cm secondary zone near the midline, 1 point for non-standard valid court areas, and 0 points for failed shots (net contact or out-of-bounds).

**Fatigue assessment measures.** Periodically assessed using the 16-point Rating of Perceived Exertion (RPE) scale (6–20 levels) [17]. Mental fatigue state: Dynamically monitored using the electronic version of the Fatigue Symptom Assessment Scale (VAFS) [18]. Data collection: RPE and VAFS data were presented in temporally synchronized formats and automatically recorded via the E-Prime 3.0 psychological experiment platform(Table 2).

## Statistical analysis

Data normality was assessed using the Shapiro–Wilk test, and homogeneity of variances was verified with Levene's test. All variables satisfied the assumptions required for parametric analyses. Data were analyzed using SPSS software (latest version, IBM Corp., Armonk, NY, USA). Descriptive statistics (mean ± standard deviation) were calculated for all outcome measures. The differences of RPE and VAFS under fatigue were calculated by single factor repeated measurement variance analysis. Post-hoc comparisons were conducted using the Bonferroni correction to adjust for multiple comparisons. Linear mixed effect model combined with Wald $\chi^2$ (which can be used to test multi-parameters) is used to test the difference of smash accuracy and smash speed under different fatigue degrees, and paired sample T test is used to compare smash speed and accuracy variables under different fatigue degrees. Multiple linear regression equation is used to fit the comprehensive fatigue degree, and the correlation between it and ball speed and accuracy is calculated by VASF and RPE as regression factors. Pearson correlation is used to calculate the correlation between smash accuracy and smash speed under different fatigue conditions. The significance level was set at $p < 0.05$.

## Results

### Temporal variations in RPE and VAFS during fatigue induction

In the process of implementing the fatigue induction paradigm, this study uses RPE scale to evaluate the physical fatigue state. At the same time, VAFS system is used to accurately evaluate mental fatigue(Tables 3 and 4).

**Table 2. Multidimensional evaluation criteria for different fatigue levels.**

|  | Indicator | Baseline | Mild fatigue | Moderate fatigue | Severe fatigue |
|---|---|---|---|---|---|
| Physical Fatigue | RPE | Score 0 | Score 1–3 | Score 4–6 | Scores ≥7 |
| Mental Fatigue | VAFS | Score 0 | Score 1–2 | Score 3–6 | Score ≥7 |

Note: RPE, Rating of Perceived Exertion; VAFS, Visual Analog Fatigue Scale.

**Table 3. Dynamic changes in fatigue-related parameters across time points (n = 24).**

|  | Baseline | Mild Fatigue | Moderate Fatigue | Severe Fatigue |
|---|---|---|---|---|
| RPE | 0.04 ± 0.10 | 2.00 ± 0.63* | 4.54 ± 0.59*# | 7.25 ± 0.60*#& |
| VAFS | 0.00 ± 0.00 | 2.21 ± 0.41* | 4.33 ± 0.64*# | 7.96 ± 0.55*#& |

Note: "*" Significant difference compared with baseline; # Significant difference for Mild Fatigue; & Compared with Moderate Fatigue, there are significant differences; RPE, Rating of Perceived Exertion; VAFS, Visual Analog Fatigue Scale.

**Table 4. Pairwise comparisons of RPE and VAFS across fatigue stages with effect sizes.**

| Measure | Comparison | Cohen's d | Interpretation |
|---|---|---|---|
| RPE | Baseline vs 20 min | 3 | Large |
| RPE | 20 min vs 40 min | 2.66 | Large |
| RPE | 40 min vs 60 min | 3.15 | Large |
| VAFS | Baseline vs 20 min | 5.32 | Very Large |
| VAFS | Baseline vs 40 min | 6.8 | Very Large |
| VAFS | Baseline vs 60 min | 14.47 | Extremely Large |

Note: RPE, Rating of Perceived Exertion; VAFS, Visual Analog Fatigue Scale.

**RPE.** Repeated-measures ANOVA revealed a significant main effect of time on RPE scores ($F(3,87) = 864.82$, $p < .001$). Post-hoc paired t-tests demonstrated statistically significant differences between all adjacent time points (all $p < .001$). Effect size analyses showed large effects between each pair of time points: from baseline to 20 minutes ($d = 3.00$), from 20 to 40 minutes ($d = 2.66$), and from 40 to 60 minutes ($d = 3.15$). These results confirm a progressive and substantial increase in perceived fatigue throughout the intervention. The fatigue degree was judged as mild after completing Block 1 and Block 2 (20 minutes), moderate after Block 3 and Block 4 (40 minutes), and severe after Block 5 and Block 6 (60 minutes).

**VAFS.** VAFS also showed a significant time-dependent effect ($F(3,87) = 1333.6$, $p < .001$). All direct comparisons between the time points and the baseline were statistically significant (Mild Fatigue – Baseline: $t(29) = 18.21$, $d = 5.32$; Moderate Fatigue – Baseline: $t(29) = 23.74$, $d = 6.80$; Severe Fatigue – Baseline: $t(29) = 28.93$, $d = 14.47$; all $p < .001$). As shown in Fig 3, the cumulative increase in VAFS scores suggests a marked acceleration of fatigue perception in the later stages of the intervention. The fatigue degree was judged as mild after completing Block 1 and Block 2 (20 minutes), moderate after Block 3 and Block 4 (40 minutes), and severe after Block 5 and Block 6 (60 minutes).

### Effects of fatigue levels on forehand smash performance in badminton

**Impact of fatigue on smash speed.** Linear mixed-effects model analysis revealed a significant effect of fatigue level on badminton Smash Speed ($\chi^2 = 32.17$, $p < .001$). Post-hoc pairwise comparisons demonstrated no significant difference between baseline and mild fatigue ($p = .145$, $d = 0.32$, small effect), but significant differences between baseline and moderate fatigue ($p = .008$, $d = 0.61$, medium effect), as well as baseline and severe fatigue ($p < .001$, $d = 0.98$, large effect). Furthermore, pairwise differences were also significant between mild fatigue and moderate fatigue ($p < .001$, $d = 1.67$), mild fatigue and severe fatigue ($p < .001$, $d = 1.41$), and moderate fatigue and severe fatigue ($p < .001$, $d = 0.96$).

Data showed no significant difference in Smash Speed performance between mild fatigue and baseline states. However, as fatigue levels increased (moderate and severe), Smash Speed demonstrated a significant and progressively larger declining trend. Notably, the magnitude of velocity decline under severe fatigue was markedly greater than that observed under mild and moderate fatigue conditions (Fig 4).

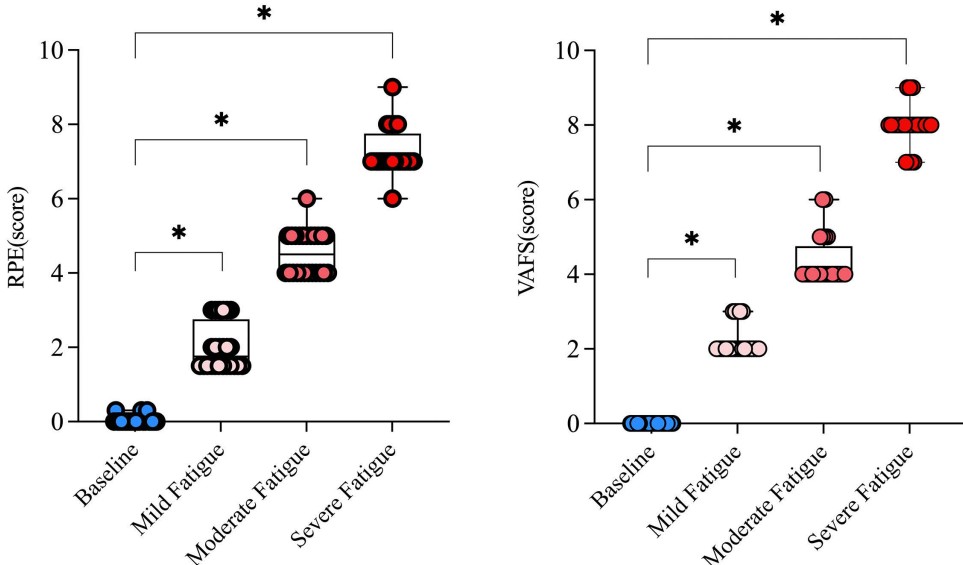

**Fig 3. Changes in RPE and VASF metrics during fatigue induction at different time points.**

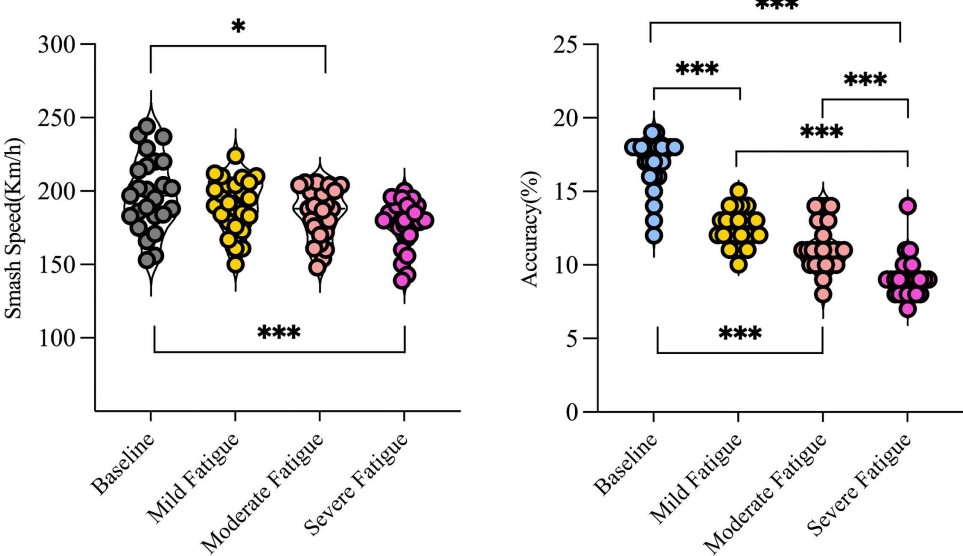

**Fig 4. Changes in smash speed and accuracy during fatigue induction at different time points.**

**Impact of fatigue on forehand smash accuracy.** Linear mixed-effects model analysis indicated a significant main effect of fatigue level on smash accuracy ($\chi^2 = 262.33$, $p < .001$). Post-hoc tests revealed that smash accuracy under moderate fatigue was significantly lower than under mild fatigue ($p = .004$, $d = 0.66$, medium effect), while all other pairwise comparisons reached statistical significance (all $p < .001$). Specifically, the accuracy decline effect sizes were substantial across comparisons: baseline vs. mild fatigue ($d = 2.09$), baseline vs. moderate fatigue ($d = 2.14$), and baseline vs. severe fatigue ($d = 3.12$), all representing large to very large effects. In addition, the comparisons between mild fatigue and severe fatigue ($d = 1.89$), and moderate fatigue and severe fatigue ($d = 1.24$) also indicated large effects.

Further analysis demonstrated a continuous and steep decline in accuracy with increasing fatigue severity, with the most pronounced difference observed under severe fatigue (Fig 4). These findings demonstrate that both Smash Speed and Accuracy deteriorate progressively as fatigue accumulates. Notably, smash accuracy demonstrates greater sensitivity to fatigue compared to Smash Speed (Tables 5–7).

## Effects of different fatigue levels on the relationship between forehand smash speed and accuracy

Pearson correlation analysis was conducted to examine the relationship between athletes' forehand Smash Speed and accuracy. Significant negative correlations were observed under moderate fatigue ($r = -0.58$, $p = .003$) and severe fatigue ($r = -0.53$, $p = .007$), whereas no significant correlations were found under baseline or mild fatigue conditions (Table 5). Differences between correlation coefficients across the four fatigue states were further compared. Results indicated a marginally significant difference between coefficients under moderate fatigue (40 min) and baseline states ($p = .086$), with no significant differences among other pairwise comparisons Table 8.

Under non-fatigued conditions, athletes can maintain high shot accuracy while executing smashes at elevated speeds. As fatigue levels increase, significant declines in smash accuracy become evident when sustaining high-speed strokes. This indicates that with progressive fatigue accumulation, athletes' skill execution progressively fails to simultaneously uphold both velocity and precision, resulting in increasingly pronounced speed-accuracy trade-offs (Fig 5).

**Table 5. Changes in various parameter metrics induced by fatigue at different time points.**

|  | Baseline | Mild Fatigue | Moderate Fatigue | Severe Fatigue |
|---|---|---|---|---|
| Smash Speed(km/h) | 198.21±25.46 | 190.71±18.91 | 184.58±17.55*# | 177.21±16.65*#& |
| Accuracy(points) | 16.92±1.78 | 12.48±1.19* | 11.32±1.65*# | 9.12±1.39*#& |

Note: "*" indicates statistically significant differences compared to baseline; "#" indicates significant differences compared to 20 minutes; "&" indicates significant differences compared to 40 minutes.

**Table 6. Pairwise comparisons of Smash Speed across fatigue levels with effect sizes.**

| Comparison | 95% CI | Cohen's d | Interpretation |
|---|---|---|---|
| Baseline vs Mild Fatigue | [-0.05, 0.89] | 0.32 | Small |
| Baseline vs Moderate Fatigue | [0.23, 1.33] | 0.61 | Medium |
| Baseline vs Severe Fatigue | [0.56, 1.89] | 0.98 | Large |
| Mild Fatigue vs Moderate Fatigue | [1.28, 3.08] | 1.67 | Large |
| Mild Fatigue vs Severe Fatigue | [0.86, 3.11] | 1.41 | Large |
| Moderate Fatigue vs Severe Fatigue | [0.48, 2.56] | 0.96 | Large |

**Table 7. Pairwise comparisons of Smash Accuracy across fatigue levels with effect sizes.**

| Comparison | 95% CI | Cohen's d | Interpretation |
|---|---|---|---|
| Baseline vs Mild Fatigue | [1.52, 3.12] | 2.09 | Large |
| Baseline vs Moderate Fatigue | [1.58, 3.26] | 2.14 | Large |
| Baseline vs Severe Fatigue | [2.27, 5.08] | 3.12 | Very Large |
| Mild Fatigue vs Moderate Fatigue | [0.27, 1.32] | 0.66 | Medium |
| Mild Fatigue vs Severe Fatigue | [1.18, 3.69] | 1.89 | Large |
| Moderate Fatigue vs Severe Fatigue | [0.61, 2.62] | 1.24 | Large |

**Table 8. Correlation analysis between Smash Speed and accuracy under different fatigue levels.**

| Fatigue State | Correlation Coefficient (r) | p-value |
|---|---|---|
| Baseline | −0.13 | 0.382 |
| Mild Fatigue | −0.21 | 0.212 |
| Moderate Fatigue | −0.58 | 0.003* |
| Severe Fatigue | −0.53 | 0.007* |

Note: *p < 0.05.

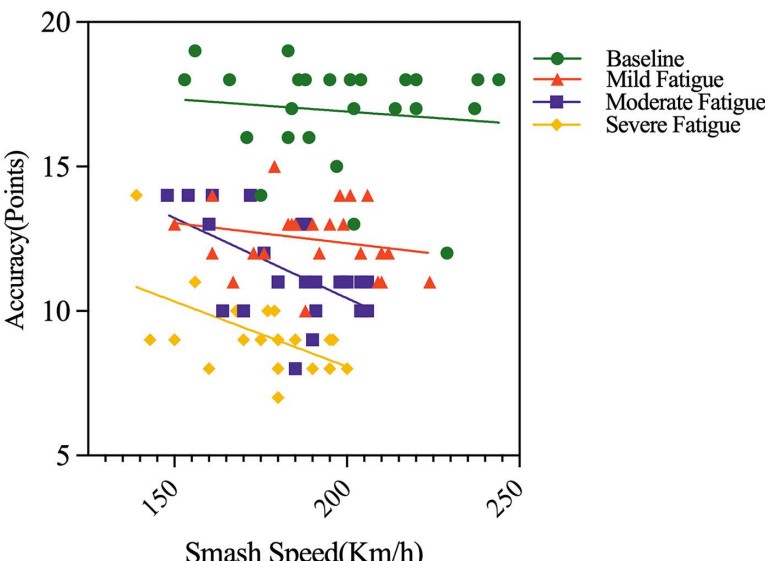

**Fig 5. Relationship between Smash Speed and Accuracy at Different Levels of Fatigue.**

## Discussion

This study, by constructing a neuromuscular-mental fatigue model, systematically investigates for the first time the synergistic effects of combined physical and mental fatigue on badminton forehand jump smash performance. The results demonstrate that as fatigue accumulates, significant declines occur in smash speed, accuracy, accompanied by the emergence of a clear speed-accuracy trade-off under moderate-to-severe fatigue. These findings highlight the compound and nonlinear impact of psycho-physiological fatigue on technical performance, offering both theoretical and practical insights into fatigue monitoring and training adaptation in high-level badminton.

This study successfully induced dual fatigue in badminton athletes. The fatigue induction protocol utilized in this investigation represents a methodological innovation compared to conventional approaches. Previous studies have typically employed 60-minute or 90-minute incongruent Stroop tasks to induce mental fatigue alone, in order to investigate its impact on badminton athletes [19,20]. Physical fatigue was induced using key exercises commonly performed by badminton athletes, including countermovement jumps (CMJs) and lunges, isokinetic knee flexion and extension tasks, as well as 18 minutes of shadow training and racket-holding drills, in order to examine the effects of physical fatigue on badminton players [8,21,22]. In comparison, the fatigue induction method adopted in this study is more specialized and closely aligned with the actual demands of badminton competition. By simultaneously inducing physical fatigue (e.g., vertical jumps, sprints) and mental fatigue (e.g., randomized target-hitting tasks), the study more accurately simulates the

complex state in which athletes endure both physical and decision-making stress in competitive scenarios. In contrast, our integrated protocol induced concurrent physical and mental fatigue through a specialized badminton-specific paradigm. By synthesizing high-intensity physical elements (vertical jumps, sprint movements) with cognitively demanding tasks (randomized target identification and hitting precision), we created a comprehensive fatigue model that authentically replicates the multidimensional stress experienced during elite competition. Although this approach does not disentangle the individual effects of each fatigue type, it reflects our deliberate focus on modeling the integrated psycho-physiological demands encountered in real-world performance contexts. This ecological approach captures the intricate interplay between physiological exertion and cognitive processing demands that characterizes high-level badminton performance.

This study demonstrates that in badminton forehand jump smashes, the synergistic effects of physical-mental mixed fatigue significantly reduce both speed and accuracy. As fatigue progresses from baseline to severe levels, smash speed decreases by 10.6% (from 198.21 km/h to 177.21 km/h), while accuracy declines by 46.1% (from 16.92 to 9.12 points). Under fatigued conditions, the peak velocity of key body segments—such as the dominant arm, forearm, and joints—significantly decreases, thereby impairing overall smash performance. These findings are consistent with previous research [23–26]. This may be attributed to fatigue-induced alterations in various kinematic variables, including reduced shoulder internal rotation, elbow extension, and wrist motion—all of which are critical for maintaining smash speed and accuracy [23,24]. Physical fatigue impairs motor unit recruitment efficiency through the accumulation of metabolic byproducts (e.g., lactate) and delayed neuromuscular transmission [4,27]. Electromyography (EMG) studies have shown that under fatigued conditions, the activation amplitude of key lower-limb muscle groups (such as the quadriceps) decreases and exhibits disordered temporal sequencing [28],resulting in insufficient power output during the take-off phase of the jump smash and subsequently reducing shuttlecock velocity. Mental fatigue may impair spatial localization and decision-making precision by suppressing metabolic activity in the prefrontal cortex (PFC) and parietal association areas [29,30]. Evidence from functional near-infrared spectroscopy (fNIRS) and functional magnetic resonance imaging (fMRI) indicates that after prolonged cognitive tasks, oxygenated hemoglobin concentration in the PFC significantly decreases ($p < 0.01$) [31], thereby reducing visual search efficiency for the shuttlecock target (e.g., fixation duration increased by 30%) [32,33]. The combined influence of physical and mental fatigue further exacerbates these impairments, leading to a greater decline in overall performance [34,35]. While our findings are behavioral in nature, the mechanistic interpretations provided are drawn from established literature in exercise neuroscience and mental fatigue.

As fatigue intensifies, it becomes increasingly difficult for athletes to maintain both high speed and high accuracy simultaneously. Under baseline or mild fatigue conditions, smash speed is almost unaffected. However, as fatigue deepens, increases in speed are often accompanied by declines in accuracy, particularly during moderate and severe fatigue stages. Physical-mental mixed fatigue affects both the physiological and psychological states of athletes, making it more challenging to sustain high accuracy at elevated speeds. The badminton smash inherently involves a decision-making process aimed at targeting specific zones on the court, and this cognitive process is subject to the speed-accuracy trade-off (SAT)—a constraint relationship in which the pursuit of higher speed typically leads to reduced accuracy, and vice versa [15]. For international-level badminton players, achieving 80–99% of their maximum smash speed represents the threshold range for attaining peak spatial accuracy [16]. Under non-fatigued or mildly fatigued conditions, individuals can rely on well-practiced skills to execute the smash effectively. However, as fatigue—especially dual fatigue—sets in, the trade-off between speed and accuracy becomes rapidly evident. Interestingly, only under moderate fatigue did the correlation between smash speed and accuracy show a marginally significant difference from the baseline, while no significant differences in correlation coefficients were found across other fatigue levels. This suggests that the speed-accuracy trade-off is most prominent during moderate fatigue, implying that athletes may need to make strategic compromises between tempo and placement precision under such conditions. Our findings align with Kahneman's Capacity Model [36], which posits that attentional resources are finite. Under moderate mixed fatigue, resource allocation becomes more constrained, leading to greater interference between cognitive and motor tasks [37,38]. This may exacerbate the speed–accuracy

trade-off observed in our participants, as both rapid decision-making and precise motor execution compete for limited capacity. This non-significant correlation observed under severe fatigue may be attributed to a floor effect, where both shot accuracy and ball speed decreased markedly, causing scores to cluster near the lower limit of the scale and reducing the variability required to detect meaningful associations [39]. Given the relatively small sample size in this study, the findings should be interpreted with caution, and future research should consider expanding the participant pool to further explore the dynamic relationship between fatigue and the speed-accuracy trade-off.

While laboratory settings control confounding variables, they fail to fully replicate competitive stressors (e.g., audience effects, real-time pressure), potentially underestimating mental fatigue impacts. Additionally, the absence of neurophysiological metrics (e.g., EEG/EMG) limits analysis of central-peripheral fatigue interactions, and the small sample size (n = 24) of national-level athletes restricts generalizability to amateur populations. Moreover, due to well-documented differences in average jump height and power output between elite male and female athletes in our recruitment pool, female participants performed a restricted-step stationary stroke instead of a jump smash to ensure technical safety and movement standardization. While this protocol limits direct biomechanical comparison between genders, it allows reliable assessment of within-subject fatigue effects. Nevertheless, future studies may further explore gender-specific fatigue responses using larger, more balanced samples. Future studies should integrate field experiments with psychophysiological monitoring (e.g., wireless surface EMG, eye-tracking) to dynamically assess muscle activation patterns and visual search strategies under real-world fatigue conditions, alongside developing personalized interventions such as cognitive-physical cross-training.

## Practical applications

The results of this study, showing significant decreases in both smash speed (moderate effect) and accuracy (large effect) following physical-mental mixed fatigue, are crucial for coaches and trainers. Since unforced errors and smash speed are key determinants of performance in badminton, it is essential for coaches and trainers to incorporate fatigue simulation into training. This approach will help minimize the negative effects of fatigue during competition, ensuring athletes can maintain performance under high-intensity conditions.

## Conclusion

This study demonstrates that combined physical and mental fatigue synergistically reduces badminton forehand jump smash performance, particularly in terms of speed and accuracy. The progressive decline in performance under increasing fatigue levels emphasizes the complex interplay between neuromuscular and cognitive systems during athletic performance. Specifically, the results show a significant decrease in both smash speed (10.6%) and accuracy (46.1%) from baseline to severe fatigue. Additionally, the emergence of a speed-accuracy trade-off, particularly under moderate and severe fatigue, highlights the challenges athletes face in maintaining both velocity and precision under fatigue.

The findings underscore the need for coaches and athletes to incorporate training strategies that address both physical and cognitive aspects of fatigue. Monitoring fatigue levels and developing interventions to mitigate its effects will be essential for optimizing performance, especially in high-stakes competitive environments. Future research should expand on these findings by incorporating neurophysiological metrics and exploring larger, more diverse athlete populations to enhance the generalizability of the results. Moreover, future studies should consider factorial designs to separate the effects of physical and mental fatigue, enabling clearer identification of their individual contributions to performance decline. Integrating cognitive-physical cross-training could also offer promising avenues for improving athletic performance in the face of combined fatigue. Additionally, researchers are encouraged to employ wearable surface electromyography (EMG) to dynamically monitor neuromuscular fatigue, and eye-tracking systems to assess visual-cognitive performance degradation under fatigue. Furthermore, incorporating match-simulated protocols may enhance ecological validity and provide deeper insights into real-world fatigue responses.

## Supporting information

**S1 File. Evaluation Data of Mixed Fatigue Levels.**
(XLSX)

**S2 File. Speed and Accuracy Under Mixed Fatigue Levels.**
(XLSX)

## Acknowledgments

The authors would like to thank Dr. Ai Jun and Dr. Liu Meng for their valuable guidance and support. Special thanks are extended to the Beijing Sport University Committee and the Jinan University Sports Committee for their assistance. The authors also express their gratitude to all the participants involved in the study for their cooperation and contribution.

## Author contributions

**Conceptualization:** Bing Wang, Lan Ding.

**Data curation:** Bing Wang, Lan Ding, Meng Liu.

**Methodology:** Yan Huang.

**Project administration:** Jun Ai.

**Supervision:** Yan Huang, Jun Ai.

**Writing – original draft:** Bing Wang, Lan Ding, Meng Liu.

**Writing – review & editing:** Meng Liu, Yan Huang.

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
