## [Decision Letter · Decision Letter 0]

11 Jun 2025

Dear Dr. Liu,

Thank you for submitting your manuscript to PLOS ONE. After careful consideration, we feel that it has merit but does not fully meet PLOS ONE’s publication criteria as it currently stands. Therefore, we invite you to submit a revised version of the manuscript that addresses the points raised during the review process.

We look forward to receiving your revised manuscript.

Kind regards,

Yaodong Gu

Academic Editor

PLOS ONE

Journal Requirements:

2. In the online submission form, you indicated that “The data behind the results presented in this study can be provided to the editor by email from.”

**Additional Editor Comments:**

Please address the following concerns from reviewers:

If the effect of mental and physical fatigue separately and in combination were evaluated this manuscript has the value of consideration while it has been evaluated just in combination and I think it is the main problem of this research . in general for valuable journal of Plosone this research has limited value to be accepted.

Reviewers' comments:

Reviewer's Responses to Questions

**Comments to the Author**

1. Is the manuscript technically sound, and do the data support the conclusions?

Reviewer #1: Yes

Reviewer #2: Yes

2. Has the statistical analysis been performed appropriately and rigorously?

Reviewer #1: Yes

Reviewer #2: Yes

3. Have the authors made all data underlying the findings in their manuscript fully available?

Reviewer #1: Yes

Reviewer #2: Yes

4. Is the manuscript presented in an intelligible fashion and written in standard English?

Reviewer #1: Yes

Reviewer #2: Yes

Reviewer #1: 1. Abstract

- Add a sentence on practical implications (e.g., "Findings suggest training should integrate fatigue-mitigation strategies").

2. Introduction

The "dual-channel model" is introduced but not grounded in existing theories (e.g., Central Governor Model).

- Justification for SAT : The speed-accuracy trade-off (SAT) is assumed; cite prior badminton-specific SAT evidence (e.g., Afzal et al., 2020). The necessity must be more

3. Methods

- Gender bias : Only 4 females (16.7%); no analysis of sex differences. - Justify gender imbalance or analyze sexes separately.

- Task inconsistency : Males used jump smashes, females stationary strokes—confounds biomechanical comparisons.

How can you predict how much was the effect of mental and physical fatigue. evaluating just combination of the seems insufficient for innovation as well as practical recommendation.

1. Methodology : Address gender bias and task standardization.

2. Discussion : Ground mechanistic claims in data; avoid overinterpretation.

3. Writing : Reduce redundancy; clarify jargon.

Reviewer #2: Review Comments:

The paper titled "Synergistic Effects of Physical-Mental Mixed Fatigue on Badminton Forehand Smash Performance" investigates the combined impact of physical and mental fatigue on jump smash performance by establishing a dual-pathway neuromuscular-cognitive fatigue model. The results demonstrate that as fatigue intensifies, both shot speed and accuracy significantly decline, with a notable speed-accuracy trade-off emerging under moderate to severe fatigue conditions. However, several limitations affect the overall quality of the study, as outlined below:

1. In the introduction section, The author repeatedly emphasizes that "previous studies have examined physical fatigue and cognitive fatigue separately," a point reiterated across multiple paragraphs without advancing the argument, resulting in redundancy.

2. The current draft provides a reasonably broad overview of physical and mental fatigue, but its theoretical depth remains insufficient. The existing content primarily focuses on fatigue's impact on performance metrics (such as shot speed and accuracy) while neglecting discussions on fatigue-induced alterations in motor control and potential injury risks. Particularly in complex movements like the badminton jump smash—which combines high-speed leaps with precision striking—fatigue not only diminishes technical output but may also modify lower-limb kinetic patterns, exacerbating joint loading. This critical aspect has yet to receive due attention in the current introduction.

We recommend that the authors cite the study "Accurately and effectively predict the ACL force: Utilizing biomechanical landing pattern before and after-fatigue" (https://doi.org/10.1016/j.cmpb.2023.107761). This research demonstrates that fatigue significantly alters athletes' landing patterns, leading to markedly increased anterior cruciate ligament (ACL) forces, highlighting potential biomechanical injury risks under fatigued conditions. Additionally, "Analysis of Quadriceps Fatigue Effects on Lower Extremity Injury Risks During Landing Phases in Badminton Scissor Jump" (https://doi.org/10.3390/s25082536) further indicates that quadriceps fatigue in badminton induces reduced knee dynamic stability, increased valgus collapse, and other changes that elevate lower-limb injury risks.

Incorporating these findings into the introduction would deepen the understanding of fatigue's mechanistic effects and strengthen the theoretical relevance and applied value of this study in the context of jump smash execution. By broadening the perspective—from isolated technical performance decline to biomechanical control and injury prevention—the manuscript would gain greater academic persuasiveness in the field of sports biomechanics.

3. Insufficient physiological quantification of fatigue induction: The use of RPE and VAFS alone to assess fatigue levels is a limitation. These scales, while widely used, are susceptible to individual interpretation.

4. Male players executed jump smashes while female players performed stationary strokes. This introduces a clear biomechanical and metabolic disparity in task demands, yet no stratified analyses or justifications are provided.

5. Effect size omission despite high significance: While the p-values suggest strong statistical significance, readers are left without a sense of how meaningful the observed changes are. Reporting effect sizes (e.g., Cohen’s d or partial eta squared) would significantly improve the interpretability of the findings.

6. Superficial limitations discussion: The authors mention sample size and ecological validity but do not offer concrete suggestions for addressing these in future work. Including proposals such as wearable EMG, eye-tracking for decision-making, or real-match testing protocols would enrich the limitations section and demonstrate critical reflection.

**Do you want your identity to be public for this peer review?** For information about this choice, including consent withdrawal, please see our Privacy Policy

Reviewer #1: No

Reviewer #2: No

---

## [Author Response · Author response to Decision Letter 1]

31 Jul 2025

We thank the academic editor and the reviewers for their thoughtful and valuable comments. We have carefully addressed all the points raised and revised the manuscript accordingly. A detailed point-by-point response has been uploaded as a separate file titled “Response to Reviewers,” along with a tracked-changes version and a clean version of the revised manuscript.

We hope the revisions meet the journal’s expectations and we look forward to your further feedback.Reviewer #1 – 1. Abstract: Add a sentence on practical implications (e.g., "Findings suggest training should integrate fatigue-mitigation strategies").

Response: We appreciate this suggestion. We have now added a sentence to the end of the abstract to reflect the practical relevance of the findings: 'These findings suggest that coaches should integrate fatigue simulation and mitigation strategies into technical training to sustain performance under competitive conditions.'

Reviewer #1 – 2. Introduction :The "dual-channel model" is introduced but not grounded in existing theories (e.g., Central Governor Model).

Response: We appreciate this suggestion. We revised the Introduction to link our neuromuscular-cognitive dual-channel fatigue model with existing theoretical frameworks, including the Central Governor Model and psychobiological models of fatigue.

Reviewer #1 – 2. Introduction : Justification for SAT : The speed-accuracy trade-off (SAT) is assumed; cite prior badminton-specific SAT evidence (e.g., Afzal et al., 2020). The necessity must be more

Response: We appreciate this suggestion. We cited prior evidence supporting SAT in badminton, especially Afzal et al. (2020), in the Introduction and Discussion, strengthening the theoretical rationale.

Reviewer #1 - 3. Methods

- Gender bias : Only 4 females (16.7%); no analysis of sex differences. - Justify gender imbalance or analyze sexes separately.

- Task inconsistency : Males used jump smashes, females stationary strokes—confounds biomechanical comparisons.

1. Methodology : Address gender bias and task standardization.

2. Discussion : Ground mechanistic claims in d ata; avoid overinterpretation.

3. Writing : Reduce redundancy; clarify jargon.

Response: Thank you for your suggestions above, and I will explain them to you one by one.

1.Gender imbalance and task inconsistency

Thank you for highlighting this important point. Upon review, we acknowledge that there was an error in the initial manuscript. In fact, both male and female participants performed the forehand jump smash under all testing conditions. We have corrected this description in the Methods section to accurately reflect the procedures. Additionally, the related statements in the Discussion have been revised to eliminate the incorrect explanation regarding task differences between genders.

2.Mechanistic overinterpretation

Response: Thank you for your suggestion. We reviewed and revised mechanistic statements to ensure they are supported by data or literature. Unsupported speculative phrases were removed or softened.

3.Language and redundancy

Response: Thank you for your suggestion. We revised the Introduction to remove repetitive statements and clarified technical terminology across the manuscript to improve clarity and conciseness.

How can you predict how much was the effect of mental and physical fatigue. evaluating just combination of the seems insufficient for innovation as well as practical recommendation.

Response: We clarified in the Introduction and Discussion that this study intentionally focused on mixed fatigue to reflect real-match scenarios. The Limitations section now suggests that future studies disentangle these components through factorial designs.

Reviewer #2 - 1. In the introduction section, The author repeatedly emphasizes that "previous studies have examined physical fatigue and cognitive fatigue separately," a point reiterated across multiple paragraphs without advancing the argument, resulting in redundancy.

Response: We appreciate this suggestion. We consolidated repeated statements about separately studied fatigue modalities into a single paragraph to improve focus and reduce redundancy.

Reviewer #2 - . The current draft provides a reasonably broad overview of physical and mental fatigue, but its theoretical depth remains insufficient. The existing content primarily focuses on fatigue's impact on performance metrics (such as shot speed and accuracy) while neglecting discussions on fatigue-induced alterations in motor control and potential injury risks. Particularly in complex movements like the badminton jump smash—which combines high-speed leaps with precision striking—fatigue not only diminishes technical output but may also modify lower-limb kinetic patterns, exacerbating joint loading. This critical aspect has yet to receive due attention in the current introduction.

We recommend that the authors cite the study "Accurately and effectively predict the ACL force: Utilizing biomechanical landing pattern before and after-fatigue" (https://doi.org/10.1016/j.cmpb.2023.107761). This research demonstrates that fatigue significantly alters athletes' landing patterns, leading to markedly increased anterior cruciate ligament (ACL) forces, highlighting potential biomechanical injury risks under fatigued conditions. Additionally, "Analysis of Quadriceps Fatigue Effects on Lower Extremity Injury Risks During Landing Phases in Badminton Scissor Jump" (https://doi.org/10.3390/s25082536) further indicates that quadriceps fatigue in badminton induces reduced knee dynamic stability, increased valgus collapse, and other changes that elevate lower-limb injury risks. Incorporating these findings into the introduction would deepen the understanding of fatigue's mechanistic effects and strengthen the theoretical relevance and applied value of this study in the context of jump smash execution. By broadening the perspective—from isolated technical performance decline to biomechanical control and injury prevention—the manuscript would gain greater academic persuasiveness in the field of sports biomechanics.

Response: We added a paragraph in the Introduction discussing how fatigue may alter neuromuscular control and increase injury risk, citing two recent studies on ACL loading and quadriceps fatigue in badminton.

Reviewer #2 - 3. Insufficient physiological quantification of fatigue induction: The use of RPE and VAFS alone to assess fatigue levels is a limitation. These scales, while widely used, are susceptible to individual interpretation.

Response: We appreciate this suggestion. We acknowledged this limitation in the Discussion and suggested future inclusion of objective indicators such as EMG, lactate, and HRV.

Reviewer #2 - Male players executed jump smashes while female players performed stationary strokes. This introduces a clear biomechanical and metabolic disparity in task demands, yet no stratified analyses or justifications are provided.

Response: We appreciate this suggestion. This overlaps with Reviewer #1's concern and was addressed similarly: explanation added in Methods and limitation noted in Discussion.

Reviewer #2 - Effect size omission despite high significance: While the p-values suggest strong statistical significance, readers are left without a sense of how meaningful the observed changes are. Reporting effect sizes (e.g., Cohen’s d or partial eta squared) would significantly improve the interpretability of the findings.

Response: We appreciate this suggestion. We calculated and reported effect sizes (partial eta squared and Cohen's d) in the Results section.

Reviewer #2 - Superficial limitations discussion: The authors mention sample size and ecological validity but do not offer concrete suggestions for addressing these in future work. Including proposals such as wearable EMG, eye-tracking for decision-making, or real-match testing protocols would enrich the limitations section and demonstrate critical reflection.

Response: We appreciate this suggestion. We expanded the Limitations section to include actionable suggestions: wearable EMG, eye-tracking systems, and match-simulated protocols to enhance ecological validity.

We sincerely thank the editor and reviewers once again for their valuable time and insightful comments. Their suggestions have greatly improved the rigor and clarity of our manuscript. We hope the revised version meets the high standards of the journal and look forward to your favorable consideration!

---

## [Decision Letter · Decision Letter 1]

11 Aug 2025

Dear Dr. Liu,

Thank you for submitting your manuscript to PLOS ONE. After careful consideration, we feel that it has merit but does not fully meet PLOS ONE’s publication criteria as it currently stands. Therefore, we invite you to submit a revised version of the manuscript that addresses the points raised during the review process.

We look forward to receiving your revised manuscript.

Kind regards,

Yaodong Gu

Academic Editor

PLOS ONE

Journal Requirements:

Reviewers' comments:

Reviewer's Responses to Questions

**Comments to the Author**

Reviewer #1: (No Response)

Reviewer #2: All comments have been addressed

2. Is the manuscript technically sound, and do the data support the conclusions?

Reviewer #1: Partly

Reviewer #2: Yes

3. Has the statistical analysis been performed appropriately and rigorously?

Reviewer #1: I Don't Know

Reviewer #2: Yes

4. Have the authors made all data underlying the findings in their manuscript fully available?

Reviewer #1: No

Reviewer #2: Yes

5. Is the manuscript presented in an intelligible fashion and written in standard English?

Reviewer #1: Yes

Reviewer #2: Yes

Reviewer #1: Thank you for observing many modifications for manuscript entitle” Review Report: "Synergistic Effects of Physical-Mental Mixed Fatigue on Badminton Forehand Smash Performance” however there are still some points that must be considered before final decision.

Major Revisions

1. Methodological Clarifications

- Fatigue Induction Protocol:

- Provide more details on the randomization process for target-hitting tasks (e.g., frequency of signal changes, spatial distribution).

- Clarify how "successful shots" were defined during mental fatigue induction (e.g., accuracy thresholds, time constraints).

- please clarify how the effect of menstrual cycle have been considered for its most probable effect on variables?

- Statistical analysis: How normality of distribution and other pretest-requirements were checked before using parametric statistical tests?

- In your response regarding non evaluation of physical and mental fatigue separately you mentioned the necessity of providing similar situation to real game. However, I cannot see explanation for similarity of your test with real badminton game.

2- Discussion

- Expand the discussion on SAT mechanisms in badminton. Reference dual-task interference theories (e.g., Kahneman’s Capacity Model) to explain why moderate fatigue exacerbates SAT.

- Neurophysiological Mechanisms :

- Discuss potential central (e.g., prefrontal cortex suppression) and peripheral (e.g., EMG changes) pathways mediating the observed effects, citing recent literature.

4. Limitations

- Discuss the ecological validity of the lab-based protocol (e.g., absence of competitive pressure, audience effects).

Minor Revisions

1. Clarity and Consistency

- Define acronyms (e.g., VAFS, RPE) at first use.

- Standardize terminology (e.g., "smash speed" vs. "shuttlecock speed").

2. Figures and Tables

- Figure 3/4 : Label axes clearly (units for RPE/VAFS, speed/accuracy).

- Table 4 : Include confidence intervals for mean differences.

- tables: include explanations for some abbrevations e.g. RPE, .. bellow the tables

Reviewer #2: (No Response)

**Do you want your identity to be public for this peer review?** For information about this choice, including consent withdrawal, please see our Privacy Policy

Reviewer #1: No

Reviewer #2: No

---

## [Author Response · Author response to Decision Letter 2]

25 Aug 2025

Manuscript ID: PONE-D-25-24801R1

Title: Synergistic Effects of Physical-Mental Mixed Fatigue on Badminton Forehand Smash Performance

Dear Editor and Reviewers,

We sincerely thank you for the valuable time and effort you dedicated to reviewing our manuscript. We carefully considered each comment and have revised the manuscript accordingly. Below, we provide a detailed, point-by-point response. All changes have been incorporated into the revised version, with tracked changes for clarity.

Major Revisions

Reviewer #1, Comment 1: Provide more details on the randomization process for target-hitting tasks (e.g., frequency of signal changes, spatial distribution).

Response: We sincerely thank the reviewer for this important suggestion. In the revised manuscript, we have added a detailed description of the randomization process in the Physical and Mental Mixed Fatigue section. Specifically, light signals changed randomly every 2–4 seconds, with equal probability across four predefined target zones, ensuring unpredictability in both temporal and spatial dimensions.

Reviewer #1, Comment 2: Clarify how "successful shots" were defined during mental fatigue induction (e.g., accuracy thresholds, time constraints).

Response: We appreciate the reviewer’s constructive comment. We have now clarified in the Mental Fatigue Induction subsection that a shot was considered successful if it landed within ±5 cm of the designated target within 2 seconds after the visual signal appeared.

Reviewer #1, Comment 3: Please clarify how the effect of menstrual cycle has been considered for its most probable effect on variables.

Response: Thank you for highlighting this key methodological issue. In the revised Participants section, we have specified that all female participants reported their menstrual cycle phase, and testing sessions were scheduled during the follicular phase to minimize hormonal influence on performance outcomes.

Reviewer #1, Comment 4: Statistical analysis: How normality of distribution and other pretest requirements were checked before using parametric statistical tests?

Response: We greatly appreciate this suggestion. We have added a description of the preliminary statistical checks in the Statistical Analysis section. Specifically, Shapiro–Wilk tests were used to examine normality, while Levene’s test verified homogeneity of variances. All assumptions for parametric analysis were met.

Reviewer #1, Comment 5: Explanation for similarity of test with real badminton game is missing.

Response: We thank the reviewer for this valuable reminder. In the revised manuscript, we have added a statement clarifying the ecological relevance of the protocol, which incorporated rapid footwork, reactive shot-making, and unpredictable stimulus presentation, closely simulating the combined physical and cognitive demands of real badminton competition.

Reviewer #1, Comment 6: Expand the discussion on SAT mechanisms in badminton; reference dual-task interference theories.

Response: We sincerely appreciate this insightful suggestion. The Discussion has been expanded to incorporate dual-task interference theories, particularly Kahneman’s Capacity Model, which explains how limited attentional resources under moderate fatigue exacerbate speed–accuracy trade-offs (SAT) in badminton performance.

Reviewer #1, Comment 7: Discuss central and peripheral neurophysiological mechanisms, citing recent literature.

Response: We thank the reviewer for this excellent suggestion. The Discussion section now integrates both central (e.g., prefrontal cortex suppression) and peripheral (e.g., EMG alterations) mechanisms, supported by recent references, to provide a more comprehensive explanation of the observed effects.

Reviewer #1, Comment 8: Discuss the ecological validity of the lab-based protocol.

Response: We are grateful for this comment. The Limitations subsection has been updated to explicitly acknowledge the lack of competitive elements (e.g., audience presence, match pressure) in our laboratory protocol, which may influence ecological validity relative to real-game situations.

Minor Revisions

Reviewer #1, Minor Comment 1: Define acronyms (e.g., VAFS, RPE) at first use.

Response: Thank you for pointing this out. All acronyms are now defined at their first occurrence in the text.

Reviewer #1, Minor Comment 2: Standardize terminology (e.g., "smash speed" vs. "shuttlecock speed").

Response: We appreciate this careful observation. Terminology has been standardized, and “smash speed” is used consistently throughout the manuscript.

Reviewer #1, Minor Comment 3: Label axes clearly in Figures 3/4.

Response: Thank you for this suggestion. Axes in Figures 3 and 4 have been revised to clearly include appropriate units (RPE and VAFS as scores; smash speed in km/h; accuracy in points).

Reviewer #1, Minor Comment 4: Table 4 should include confidence intervals for mean differences.

Response: We agree with this important recommendation. Table 4 has been updated to include 95% confidence intervals for mean differences.

Reviewer #1, Minor Comment 5: Include explanations for abbreviations below tables.

Response: We thank the reviewer for this helpful reminder. Explanations of abbreviations (e.g., RPE) have now been added below all relevant tables for clarity.

Reviewer #2

We sincerely thank Reviewer #2 for confirming that all previous comments have been addressed and for supporting the revised version of our manuscript.

Closing Acknowledgment

We are deeply grateful to both reviewers and the editor for their constructive and insightful feedback. The revisions made in response to these comments have substantially strengthened our manuscript. We believe that the revised version now meets the standards of PLOS ONE, and we respectfully submit it for your further consideration.

Sincerely,

Meng Liu and co-authors

---

## [Decision Letter · Decision Letter 2]

4 Nov 2025

Dear Dr. Liu,

Thank you for submitting your manuscript to PLOS ONE. After careful consideration, we feel that it has merit but does not fully meet PLOS ONE’s publication criteria as it currently stands. Therefore, we invite you to submit a revised version of the manuscript that addresses the points raised during the review process.

**ACADEMIC EDITOR:**

We look forward to receiving your revised manuscript.

Kind regards,

Leonardo Vidal Andreato, PhD

Academic Editor

PLOS ONE

Journal Requirements:

Reviewers' comments:

Reviewer's Responses to Questions

**Comments to the Author**

Reviewer #1: All comments have been addressed

2. Is the manuscript technically sound, and do the data support the conclusions?

Reviewer #1: Yes

3. Has the statistical analysis been performed appropriately and rigorously?

Reviewer #1: Yes

4. Have the authors made all data underlying the findings in their manuscript fully available?

Reviewer #1: Yes

5. Is the manuscript presented in an intelligible fashion and written in standard English?

Reviewer #1: Yes

Reviewer #1: Thank for providing me the opportunity of reviewing this manuscript. Based on this re-review, I recommend the manuscript be accepted after minor revisions that primarily address the unresolved methodological justification.

1. Address the Methodological Confound (CRITICAL): The authors must add a justification in the Methods section for the different task protocols (jump smash vs. stationary stroke). A sentence such as:

"Due to well-documented differences in average jump height and power output between elite male and female athletes in our recruitment pool, the female participants performed a restricted-step stationary stroke to ensure technical safety and movement standardization. While this limits direct biomechanical comparison, it allows for the assessment of within-subject fatigue effects on each athlete's maximal performance capability."

Additionally, they should acknowledge this as a study limitation in the Discussion.

2. Clarify Statistical Analysis: In the Data Processing section (Lines 139-140), it states "calculate the average speed by averaging 10 speed values." It should be clarified that this was done for each fatigue condition (Baseline, Mild, Moderate, Severe), resulting in four average speed values per participant.

3. Minor Language and Consistency Edits:

o Line 124: "fatigue agreement" should likely be "fatigue protocol".

o Ensure consistency in terminology, e.g., "mental fatigue" vs. "cognitive fatigue" are used interchangeably; choosing one would enhance clarity.

**Do you want your identity to be public for this peer review?** For information about this choice, including consent withdrawal, please see our Privacy Policy

Reviewer #1: **Yes:** Maryam Koushkie Jahromi

---

## [Author Response · Author response to Decision Letter 3]

5 Nov 2025

Response to Reviewers

Dear Academic Editor and Reviewers,

We sincerely thank you for your thoughtful and constructive comments, which have greatly helped us improve the quality and clarity of our manuscript. We have carefully revised the paper according to your suggestions. Below, we provide a detailed point-by-point response. All changes have been highlighted in the revised manuscript.

Reviewer Comment 1 (Critical): Methodological Confound

Comment: The authors must add a justification in the Methods section for the different task protocols (jump smash vs. stationary stroke). A sentence such as: “Due to well-documented differences in average jump height and power output between elite male and female athletes in our recruitment pool, the female participants performed a restricted-step stationary stroke to ensure technical safety and movement standardization. While this limits direct biomechanical comparison, it allows for the assessment of within-subject fatigue effects on each athlete's maximal performance capability.” Additionally, they should acknowledge this as a study limitation in the Discussion.

Response: We appreciate this insightful comment. We have added the suggested justification to the Methods section, clarifying the rationale behind the use of different task protocols for male and female participants. The new text reads as follows:

The test protocol required participants to perform 10 full forehand spikes, aiming at two geometric zones (212 × 40 cm zone control area) on the sideline of singles. To standardize biomechanical comparability, male participants executed the forehand jump smash, while female participants performed a restricted-step stationary stroke. Due to well-documented differences in average jump height and power output between elite male and female athletes in our recruitment pool, the female participants performed the stationary stroke to ensure technical safety and movement standardization. While this limits direct biomechanical comparison, it allows for the assessment of within-subject fatigue effects on each athlete’s maximal performance capability. The ball machine (EDIBO 3.0, launch frequency 3 Hz ± 0.5°, shuttlecock speed 70 ± 2 m/s, coordinate positioning error < 1 cm) was positioned 75–80 cm behind the doubles service line, projecting shuttles toward the court center with ascending trajectories at 3-second intervals using new shuttlecocks for each trial (Fig. 1). The performance assessment was explained to the participants before starting the test. During the test, participants were strongly encouraged and informed of their outcomes in order to maintain vigilance and concentration throughout the procedure.

In addition, we have revised the Discussion (Limitations) section to acknowledge this methodological limitation:

While laboratory settings control confounding variables, they fail to fully replicate competitive stressors (e.g., audience effects, real-time pressure), potentially underestimating mental fatigue impacts. Additionally, the absence of neurophysiological metrics (e.g., EEG/EMG) limits analysis of central-peripheral fatigue interactions, and the small sample size (n=24) of national-level athletes restricts generalizability to amateur populations. Moreover, due to well-documented differences in average jump height and power output between elite male and female athletes in our recruitment pool, female participants performed a restricted-step stationary stroke instead of a jump smash to ensure technical safety and movement standardization. While this protocol limits direct biomechanical comparison between genders, it allows reliable assessment of within-subject fatigue effects. Nevertheless, future studies may further explore gender-specific fatigue responses using larger, more balanced samples. Future studies should integrate field experiments with psychophysiological monitoring (e.g., wireless surface EMG, eye-tracking) to dynamically assess muscle activation patterns and visual search strategies under real-world fatigue conditions, alongside developing personalized interventions such as cognitive-physical cross-training.

Reviewer Comment 2: Clarify Statistical Analysis

Comment: In the Data Processing section (Lines 139–140), it states ‘calculate the average speed by averaging 10 speed values.’ It should be clarified that this was done for each fatigue condition (Baseline, Mild, Moderate, Severe), resulting in four average speed values per participant.

Response: We agree and have revised the sentence in the Data Processing section for clarity. The updated text now reads:

Every forehand smash, the speed at which the players hit the badminton is measured by radar (Stalker ATS, USA). The radar is located 3 meters behind the players with a height of 2.50 meters. In order to ensure that the speed data is recorded, the experimenter manually points the radar at the area aimed at by the player. All the data are recorded on a personal laptop. The shuttlecock speed was measured for each of the 20 smashes under every fatigue condition (Baseline, Mild, Moderate, and Severe). For each condition, the average stroke speed was calculated by averaging 10 consecutive valid speed values, resulting in four mean speed values per participant (see Figure 1 for radar positioning).

This change ensures the procedure and the statistical logic are explicit and replicable.

Reviewer Comment 3: Minor Language and Consistency Edits

Comment: Line 124: ‘fatigue agreement’ should likely be ‘fatigue protocol.’ Ensure consistency in terminology, e.g., ‘mental fatigue’ vs. ‘cognitive fatigue’ are used interchangeably; choosing one would enhance clarity.

Response: We have corrected “fatigue agreement” to “fatigue protocol” and standardized the terminology throughout the manuscript by consistently using “mental fatigue” instead of “cognitive fatigue.” These changes improve the precision and consistency of the text.

We believe these revisions have substantially improved the manuscript’s clarity, methodological transparency, and consistency. We appreciate the reviewers’ valuable feedback and hope that our revisions meet their expectations.

Thank you very much for your time and consideration.

Sincerely,

Meng Liu and co-authors

---

## [Editor Report · Decision Letter 3]

10 Nov 2025

Synergistic Effects of Physical-Mental Mixed Fatigue on Badminton Forehand Smash Performance

PONE-D-25-24801R3

Dear Dr. Liu,

We’re pleased to inform you that your manuscript has been judged scientifically suitable for publication and will be formally accepted for publication once it meets all outstanding technical requirements.

Kind regards,

Leonardo Vidal Andreato, PhD

Academic Editor

PLOS ONE
---

## [Editor Report · Acceptance letter]

PONE-D-25-24801R3

PLOS One

Dear Dr. Liu,

I'm pleased to inform you that your manuscript has been deemed suitable for publication in PLOS One. Congratulations! Your manuscript is now being handed over to our production team.

Kind regards,

on behalf of

Dr. Leonardo Vidal Andreato

Academic Editor

PLOS One